# Oxidation Behavior Characterization of Zircaloy-4 Cladding with Different Hydrogen Concentrations at 500–800 °C in an Ambient Atmosphere

**DOI:** 10.3390/ma15092997

**Published:** 2022-04-20

**Authors:** Tung-Yuan Yung, Wen-Fang Lu, Kun-Chao Tsai, Cheng-Hao Chuang, Po-Tuan Chen

**Affiliations:** 1Institute of Nuclear Energy Research, Lungtan, Taoyuan 32546, Taiwan; tyyung@iner.gov.tw (T.-Y.Y.); wflu@iner.gov.tw (W.-F.L.); tsaijohn@iner.gov.tw (K.-C.T.); 2Department of Physics, Tamkang University, New Taipei 251, Taiwan; 3Department of Vehicle Engineering, National Taipei University of Technology, Taipei 106, Taiwan

**Keywords:** zircaloy-4 cladding, oxidation kinetics, zirconium hydride, weight gain

## Abstract

This study simulated the after-burned zirconium cladding oxidation in air at temperatures between 500 and 800 °C. The weight changes of Zircaloy-4 cladding with hydrogen contents of 100–1000 ppm continuously measured through thermogravimetric analysis (TGA) during oxidation tests at different temperatures in an air atmosphere. The TGA results indicate a transition of oxidation kinetics from a parabolic rate law to a linear rate law for as-received and hydrided Zircaloy-4 cladding. The hydrogen concentration of Zircaloy-4 had a marked effect on its pre-transition oxidation in air between 500 and 800 °C. For all samples, the linear oxidation (post transition stage) at 650 °C, which is the critical oxidation temperature, displays a similar trend. In addition, scanning electron microscopy and transmission electron micros-copy examinations indicated the presence of a few and numerous discontinuous micro-cracks in the oxide layer in the pre-transition and post-transition stages, respectively.

## 1. Introduction

In nuclear power plants, severe accidents, such as the breach of a reactor pressure vessel or the loss of spent fuel pool cooling water, result in exposure of fuel rods to high-temperature steam and air with excessive oxidation of the Zr alloy fuel cladding [1]. Moreover, the structural failure of a dry storage canister in a nuclear power plant results in air entering the canister and subsequently interacting with the spent fuel rods [2]. Zirconium and its alloys are easily oxidizing because of their affinity to oxygen [3,4]. When Zircaloy fuel cladding is exposed to air and steam environments at elevated temperatures [5], zirconium oxide is produced. As the spent fuel zirconium claddings after burn-up, 100–1000 ppm hydrogen contents are observed.

The integrity of fuel rods and the oxidation kinetics of Zircaloy cladding in high-temperature environments have significantly investigated. Studies have examined the high-temperature oxidation behavior of Zircaloy cladding in air and steam atmospheres [6,7,8,9]. Several studies have evaluated the oxidation kinetics of Zircaloy in steam environments; however, most of these studies were performed at temperatures higher than 700 °C on bare zirconium alloys for short periods to predict the cladding oxidation behavior under a loss-of-coolant-accident (LOCA) condition [10,11,12,13]. Few studies have investigated the long-term oxidation behaviors of Zircaloy cladding in air at temperatures below 700 °C [13]. Furthermore, fewer studies have focused on the effect of hydride on the oxidation kinetics of zirconium alloys. 

Courty et al. reported hydride precipitation kinetics in Zircaloy-4 studied with synchrotron X-ray diffraction. They simulated the in-reactor corrosion during operation in nuclear reactors at 288–400 °C. Their results showed that average hydride precipitation rate was no dependent on the initial concentration of hydrogen [14]. To Ref. [5], the zircaloy-4 oxidation in stream atmosphere could hydrogen content up to ~600 ppm in 600–1200 °C. Since the reaction enthalpy of zirconium alloy in air is higher than in steam (−1100 kJ/mol vs. −616 kJ/mol), the degradation of Zircaloy-4 would be faster in the air atmosphere than in the steam. The metal is easily embrittled with the hydrogen uptake occurring. The mechanical properties would be weakened with metal hydride in the oxide layer. During metal in corrosion chemical reaction, the hydrogen is releasing as the reaction product. Hydrogen easily diffuses into zirconium oxide layer to form the zirconium hydride when the hydrogen concentration is solid solubility limit. Furthermore, the zirconium hydride reduces the ductility of cladding and makes cracks on the oxide layers as reported. 

The present study evaluated the effect of hydride on the oxidation behavior of Zircaloy-4 in air atmosphere. The oxidation kinetics of as-received and hydrided Zircaloy-4 cladding specimens with hydrogen concentrations between 100 and 1000 ppm was investigated in detail in the temperature range of 500–800 °C. The Zr hydride could be formed around 180~280 °C with certain amount of hydrogen. The typical temperature is below 370 °C for spent fuels in pressurized water reactors (PWR) after 5-year service and minimum dry storage for 20 years. One of the off normal temperature conditions in short-term is below 570 °C. Thus, the experiment temperature range in this study is set at 500–800 °C for the normal and off normal storage scenarios [15]. Extensive metallographic examinations of cross sections of the pre-hydrided and air-oxidized specimens were conducted to determine the oxide thickness and hydride distribution across the cladding wall. After oxidation, the hydrogen concentrations of the air-oxidized specimens were determined. The hydrogen concentration results were used to correlate the parabolic and linear rate constants with the hydrogen concentration of the specimens. In addition, the microstructures of the oxide scales formed after oxidation at 650 °C were characterized through field-emission scanning electron microscopy (FE-SEM) and transmission electron microscopy (TEM). Electron-energy loss spectroscopy (EELS) was used to analyze the oxidation layer composition. The crystalline structures of the as-received Zicaloy-4 and Zicaloy-4 with a hydrogen concentration of 750 ppm analyzed through X-ray diffraction (XRD).

## 2. Experimental Methods

### 2.1. Materials and Hydrogen Charging Process

The as-received Zircaloy-4 cladding tubes used in this study were stress relief–annealed at 596 °C for 3.5–4.5 h. The outside diameter and wall thickness of the Zircaloy-4 cladding were 9.5 and 0.58 mm, respectively. The chemical composition of the aforementioned cladding is listed in Table 1. To simulate the properties of spent fuel cladding, the Zircaloy-4 cladding tubes were first hydrogen-charged to the target hydrogen concentrations of 120, 350, and 750 ppm. Then, 13-cm-long specimens were cut from the Zircaloy-4 cladding for hydridation and hydrogen-charged through thermal cycling. These specimens encapsulated with a predetermined amount of pure hydrogen gas in a Pyrex capsule (Corning Inc., New York, NY, USA). This capsule has a sufficient volume to maintain low partial pressure of hydrogen to prevent the formation of hydride layers. Depending on the target hydrogen concentration, the encapsulated cladding specimens were thermally cycled between 200 and 300 °C for a certain number of cycles with heating and cooling rates of approximately 3 and 2 °C/min, respectively. The hydrogen concentrations of the hydrided Zircaloy-4 were determined by using a hydrogen detector (TCH600 Hydrogen Determinator, Leco Inc., St. Joseph, MI, USA).

### 2.2. High-Temperature Oxidation

The specimens for the high-temperature oxidation tests were cut from the pre-hydrided cladding tubes. These specimens had a length of approximately 10 mm, and their top and bottom were polished using emery papers with grit sizes of up to 2500 μm. High-temperature oxidation tests were conducted on Zircaloy-4 cladding by using a thermogravimetric analyzer (TGA Sestsys EVO, Setaram Inc., Caluire-et-Cuire, France). A symmetrical hang-down balance used to record the weight changes of the cladding continually with a ±0.1-μg resolution during the oxidation tests. The temperatures of the samples were controlled and measured accurately within ±0.1 °C in the oxidation test. Argon (Ar) gas purged into the heating chamber at a flow rate of 20 sccm to protect the heating device. The heating rate to the target temperatures of 500–650 °C was set as 20 and 30 °C/min for testing at 700 and 800 °C, respectively. The exposure times for testing at 500, 550, 600, 650, 700 and 800 °C were 150, 96, 50, 50, 10 and 3 h, respectively. The oxidized samples were removed from the heating chamber when they cooled to room temperature. 

### 2.3. Metallographic Examinations and Crystalline Characterization

Metallographic examinations were conducted to characterize the hydride distributions of the hydrided cladding specimens. For the aforementioned examinations, the hydrided cladding specimens were prepared according to the standard metallographic preparation procedures and etched with an etching solution; the HF:HNO_3_:H_2_SO_4_:H_2_O ratio was 1:10:10:10. A laser optical system (LEXT OLS4100, Olympus Corporation, Tokyo, Japan) was employed to measure the oxide thickness of the as-received and hydrided Zircaloy-4 specimens and to investigate their metallographic features. The oxide morphology was further examined using a FE-SEM (JOEL JEM 6700, JEOL Ltd., Tokyo, Japan) and a TEM equipped with an EELS (JOEL FEM ARM 200F, JEOL Ltd., Tokyo, Japan) for determining the oxide composition of the specimens. A dual column ultra-high-resolution FE-SEM (FEI Nova 200, Thermo Fisher Scientific Inc., Waltham, MA, USA) on the TEM specimens, which had dimensions of 4 mm × 2 mm and a thickness of 25–30 nm. The oxides of the specimens were examined using an X-ray diffractometer with a Cu Kα X-ray source (D8 diffractometer, Bruker AXS, Karlsruhe, Germany). The phase content was analyzed with Bruker EVA 4.2 software (Bruker AXS, Karlsruhe, Germany) in semi-quantification method. 

## 3. Results and Discussion

Figure 1 shows the optical micrographs of the as-received and hydrided Zircaloy-4 specimens. Zirconium hydrides were oriented in the circumferential direction and distributed homogeneously across the cross sections of the cladding samples, as displayed in Figure 1b–d. The space between the hydride precipitates decreased with an increase in hydrogen concentration. 

Figure 2 depicts the surfaces of the as-received Zircaloy-4 specimens after oxidation at various temperatures. The surfaces of the specimens exhibited different colors depending on the thickness of the oxide layer. The surface of the specimen oxidized at 500 °C was black. The surface then turned light brown when the oxidation temperature increased to 550, 600, 650, and 700 °C. At 800 °C, some white spots were observed on the brown specimen surface. Moreover, in terms of their shade of brown, color and glossiness variations were observed on the surfaces of the cladding specimens oxidized between 500 and 800 °C. This variation in color shade might been caused by the rate transition during oxidation testing. Rate transition can affect the oxidation kinetics or increase the oxidation rate at different test temperatures. Baek et al. observed changes in the colors of ZIRLO™ and Zircaloy-4 surfaces at test temperatures of 700–1200 °C [10]. Furthermore, the Zircaloy-4 would shorten the breakaway time as temperature increasing, because the oxidation rate constants of the Zircaloy-4 were strongly dependent on the breakaway oxidation kinetics. 

Figure 3 depicts the TGA weight gain curves for the as-received and hydrided Zircaloy-4 cladding specimens in the temperature range of 500–800 °C. Most of the weight gain curves for Zircaloy-4 exhibit a transition from a parabolic rate law (pre-transition) to a linear rate law (post-transition). The weight gain data collected through TGA the amount of oxygen and nitrogen absorbed during the oxidation test. Two of possible oxidation reactions of Zr under operation conditions (ambient atmosphere with average relative moisture ~70% in Taiwan) are expressing as followings:Zr + O_2_ → ZrO_2_, −1100 KJ/mol at 298 K (1)
Zr + 2H_2_O → ZrO_2_ + H_2_, −616 KJ/mol at 298 K (2)

Steinbrueck et al. [5] reported that nitrogen has strong effects on the transition times and oxidation kinetics of Zircaloy. They performed Raman spectroscopy with an X–Y stage as well as image analysis and determined that the presence of nitrogen may result in the formation of Zr derivatives containing ZrO_2_, ZrN, or ZrO_x_N_y_ at temperatures above 800 °C.

The crystalline surfaces of the specimens with pre-transition kinetics characterized through XRD in this study. Figure 4 displays the XRD spectra of the as-received Zircaloy-4 and Zircaloy-4 with a hydrogen concentration of 750 ppm with 300 °C thermal cycling treatment. The as-received Zircaloy-4 exhibited peaks of crystalline zirconium at 2θ values of 32.1°, 34.2°, 36.3°, 47.9°, and 63.4° in the crystallography open database (COD) database analysis. The hydrided Zircaloy-4 with 750 ppm of hydrogen exhibited a broad Zr oxide peak (similar to the as-received Zircaloy-4), Zr hydride peaks (at 2θ = 37.7°, 54.2° and 64.7°), and Zr oxide peaks (at 2θ = 32.0°, 34.3°, 36.4°, 47.7°, and 63.3°) before oxidation at different temperatures in an air atmosphere. With Bruker EVA 4.2 semi-quantified method analyzing for phase content on the surface, the Zr:Zr(O):ZrH ratio is about 72.1%:22.7%:5.2%, respectively.

In most studies on the high-temperature oxidation of Zircaloy, it has been assumed that the thickness of an oxide after oxidation increases with the square root of oxidation time in the early stage [16,17,18,19,20,21,22]. This variation called the parabolic rate law and results from the diffusion of oxygen through the oxide layer according to the parabolic reaction rate. For a parabolic reaction rate, the relationship between the weight gain and the exposure time is as following:∆*W*^2^ = *K_p_* · *t* + *C*(3)
where *W* is the weight gain at the surface of the specimen, *K_p_* is the parabolic rate constant, *t* is the exposure time, and *C* is a constant. 

When the oxidation kinetics are in the linear rate region, the relationship between the weight gain and the exposure time is as follows:∆*W*^2^ = *K_L_* · *t* + *C*(4)
where *K_L_* is the linear rate constant. 

Most of the specimens oxidized below 800 °C exhibited linear kinetics. The transitions of the weight gain curves caused by the formation of cracks in the oxide layer (which is called breakaway). These cracks allowed oxygen or other oxidants to come into contact with the metal, which resulted in a transition from a parabolic rate law to a linear rate law. Leistikow et al. observed a similar transition phenomenon during the oxidation of Zircaloy [23]. With the progression of oxidation in the linear region, a porous ZrO_2_ layer with long cracks formed. The microstructural features of the oxide layer is described below.

The transition time of weight gain curves depended on the test temperature and the hydrogen concentration of the relevant Zircaloy-4 cladding specimen. The weight gain curves of all the specimens exhibited the same trend in the parabolic rate region at all the test temperatures. The shortest breakaway time for the oxidation at the maximum test temperature of 800 °C was approximately 35–40 min, and the shortest breakaway time varied marginally with hydrogen concentration. Although all the specimens oxidized under the same parabolic rate law at all the test temperatures, their weight gain rates after breakaway were marginally different due to the variations in their hydrogen concentrations. Lasserre et al. [24] reported the oxygen and nitrogen mixture atmospheres could also observe same phenomenon as abovementioned. In particular, all the weight gain curves for 650 °C consistent before the change from a parabolic rate law into a linear rate law (as indicated by the arrows for Zircaloy-4 as-received data in Figure 3). During the pre-transition period, a protective ZrO2 scale is formed (and oxidation determined by diffusion of O through this growing scale). Various mechanics may result in the formation of cracks (breakaway). After transition, the oxidation kinetics is determined by the diffusion of the oxidizing gas through these cracks to the metal/oxide interface [5]. The duplication of the weight gain curves in a periodic manner could attribute to the reaction occurring at the metal interface. 

In this study, the weight gains were measured under a parabolic rate law (pre-transition stage) and linear rate law (post-transition stage) [25,26]. Table 2 lists the parabolic and linear oxidation rate constants for the as-received and hydrided Zircaloy-4 alloys which are calculated from the weight gain curves in Figure 3. In the pre-transition stage, the highest oxidation rate constants (of approximately 10^−10^ kg/m^2^·s) are obtained for the oxidation of the hydrided Zircaloy-4 specimen with 750 ppm of H at 500, 550 and 800 °C. For oxidation at 600–700 °C, the hydrogen concentrations had no significant effect on the oxidation rate constant. In the post-transition stage, the oxidation rate constants of the as-received and hydrided Zircaloy-4 specimens were calculated to be approximately 10^−8^ Kg/m^2^·s at 500 °C and approximately 10^−5^ Kg/m^2^·s at 800 °C. The calculated rate constants indicate that the oxidation of the Zircaloy-4 cladding specimens was strongly dependent on temperature. The parabolic and linear rate constants of these specimens expressed as follows:*K_p_* or *K_L_* = *A* exp(−*Q*/R*T*)(5)
where *K_p_* and *K_L_* are the parabolic and linear rate constants, respectively; *A* is a constant; *Q* is the activation energy for the oxidation reaction; *R* is the universal gas constant (8.314 J/mol K); and *T* is the oxidation temperature (K).

By fitting the oxidation rate constant and temperature curves with the Arrhenius equation, the rate constant equations for *K_p_* and *K_L_* can be obtained (Table 3, respectively). From these equations, the ∆*W* value during the oxidation period at a specific temperature could be estimated. From Table 3 and Figure 5c, the activation energies (*Q*) of the oxidation region based on the parabolic rate law for the Zircaloy-4 specimens with 750, 350, and 120 ppm of H were obtained as 39.3, 31.5, and 35.5 kcal/mol, respectively. An increase in the hydrogen content increased the activation energy of the pre-transition kinetics. The activation energies of the post-transition kinetics for the Zircaloy-4 specimens with 750, 350, and 120 ppm of H were 37.5, 38.6, and 36.1 kcal/mol, respectively. The activation energies of the pre-transition and post-transition kinetics for the as-received Zrcaloy-4 were 38.7 and 36.1 kcal/mol, respectively. A parabolic rate law and short oxidation time were achieved easily for the oxidation of the Zircaloy-4 specimens with a hydrogen concentration of less than 750 ppm. Moreover, the same linear-rate oxidation behaviors are observed for the hydrided and as-received specimens. In a previous study, the oxidation activation energies of Zircaloy-4 were approximately 19.97 kcal/mol at 1050–1500 °C in an air atmosphere, approximately 20.83 kcal/mol at 1000–1300 °C in an air atmosphere, and approximately 33.4 kcal/mol in a stream environment at 841–1482 °C [27]. The activation energies for oxidation based on the parabolic rate law (pre-transition stage) at 500–800 °C in the current study are higher than the aforementioned activation energy values. Cubic oxidation only was observed for the specimen with 750 ppm of H at 550 °C. Moreover, the as-received Zrcaloy-4 specimen exhibited cubic kinetic phenomena.

However, the hydrogen content did not affect the activation energy of the post-transition kinetics [Table 3 and Figure 5d]. Nevertheless, the analyzed hydrogen contents for this study, the 750 ppm hydrogen charging specimens are more than 750 ppm and more than 1000 ppm, shown in Table 2. Excluded the 750 ppm specimens, the rate constant of parabolic region is increasing with increasing hydrogen contents. The rate constant of linear region seems not affected by hydrogen contents and the highest rate constant is 350 ppm hydrogen content only.

Figure 5a depicts the parabolic rate constants of the Zircaloy-4 specimens oxidized in the pre-transition stage as functions of their hydrogen concentration. A plot of the linear rate constants of the Zircaloy-4 specimens oxidized in the post-transition stage versus their hydrogen concentration is displayed in Figure 5b. The aforementioned figures indicate that the hydrogen concentration (100–1000 ppm) had significant effects on the oxidation kinetics in air at 500–800 °C. Moreover, some minor fluctuations were observed in the parabolic and linear rate constants with different hydrogen concentrations at 800 °C, shown in Figure 5. A quantitative dependence between the rate constant and hydrogen concentration cannot be determined because of the minor variations in the rate constant with hydrogen concentration at 800 °C. In a study conducted by Argonne National Laboratory, the oxidation kinetics of Zirlo in air exhibited a negligible dependence on hydrogen concentrations between 100 and 1000 ppm at test temperatures of 300–600 °C [28]. Thus, the oxidation behavior of Zircaloy-4 may be unaffected by hydrogen concentration at 650 °C in the post-transition stage. 

As Grosse, M. [29] et al. published results about zircaloy-4 from 600–1200 °C with various nitrogen/stream atmosphere, the highest hydrogen uptake was up to 3600 ppm. The oxidation behaviors were from parabolic to linear stages during incubation time. The oxide layer of zircaloy-4 formed as the parallel-like porous structure losing the protective capability. Although our oxidation temperature is at 500–800 °C range, our finding on oxidation at 650 °C in air temperature show almost same trend in linear region (post-transition stage). Furthermore, the rate constant of linear region at 650 °C is highest value with various hydrogen contents. 

Figure 6 presents the experimental and calculated ∆*W* values for the as-received Zircaloy-4 after oxidation at temperatures of 500, 550, 600, 650, 700 and 800 °C for exposure times of 150, 96, 50, 50, 10 and 3 h, respectively. Since the oxidation kinetics of the Zircaloy-4 cladding may be affected by hydrogen concentration (the 750 ppm hydrogen charging method was not consistence with calculated hydrogen content in this study, shown in Table 2), the ∆*W* value for the as-received Zircaloy was estimated as an example to determine the variations in the weight gain with an increase in oxidation temperature significantly. The largest weight gain at 650 °C obtained for a longer exposure time of 50 h, it is about 1800 mg/dm^2^. Furthermore, the obtained experimental and calculated weight gains were similar for each oxidation temperature. Thus, the weight gain was estimated from the calculated rate constants at a specific temperature.

The microstructures of the oxide scale of the as-received Zircaloy-4 specimen oxidized at 650 °C for 1, 10, and 50 h are shown in Figure 7. Figure 7a displays the microstructure of the as-received Zircaloy-4 specimen after oxidation at 650 °C for 1 h. The oxide layer had a dense and intact structure in the pre-transition stage. Although the oxide scale had some small cracks, it still functioned as a protective layer for the metal. The thickness of the oxide layer was approximately 5 μm after oxidation at 650 °C for 1 h. Figure 7b,c displays the microstructures of the oxide scale of the as-received specimen oxidized at 650 °C for 10 and 50 h, respectively. Both these microstructures exhibited characteristics of an oxide layer in the post-transition region. Continuous cracks observed across the oxide scale. The formation of a loose structure of cracks and pores implied that the oxidation kinetics of Zircaloy-4 followed the linear rate law. The thickness of the oxide scale was approximately 25 μm after oxidation for 10 h. 

The Zircaly-4 specimen with 350 ppm of hydrogen was subjected to TEM analysis after its oxidation at 650 °C for 25 h. Figure 8a,b depicts the cross sections of the specimen with the Zr substrate and Zr oxides as well as the selected area electron diffraction patterns for the Zr substrate and Zr oxides. The Scanning TEM image used for EELS analysis is in Figure 8c, and the EELS mapping of Zr and O is illustrated in Figure 8d. EELS element mapping indicated that after oxidation, the aforementioned specimen was dominated by Zr and O but did not contain zirconium nitride maybe the thicknesses of the transition layer and transition zone were approximately 25–30 and 50 nm, respectively. There is a report for nitride acts as the “catalyst” in zirconium oxidation [5]. This shows that the parabolic rate region is approximately 5 h with the thickness of the transition zone. In this study, the oxidation of Zircaly-4 specimens in an air atmosphere followed the parabolic rate law in the pre-transition stage and the linear rate law in the post-transition stage.

## 4. Conclusions

In this study, with simulated after-burned cladding, the hydrided Zircaloy-4 specimens, were oxidized at 500–800 °C. By fitting the oxidation rate constants and temperature data with the Arrhenius equation, the rate constant equations for *K_p_* and *K_L_* were obtained in two stages, parabolic and linear oxidation stages. By using these equations and the calculated rate constants, ∆*W* is obtained for a specific temperature. The rate constant data indicated that the hydrogen concentration of Zircaloy-4 (in the range of 100–1000 ppm) significantly affected the pre-transition oxidation in air at 500–800 °C. An increase in the hydrogen content increased the activation energy of the pre-transition kinetics. Zircaloy-4 specimens with 750, 350, and 120 ppm of H obtained the oxidation activation energies as 39.3, 31.5, and 35.5 kcal/mol, respectively. The oxide layers had a dense and intact structure in the parabolic rate region. The formation of continuous cracks and pores in the oxide scale implied that the oxidation kinetics of Zircaloy-4 followed the linear rate law. The oxide scales that formed after oxidation examined through XRD and TEM to determine the linear and parabolic kinetic behaviors of Zircaloy-4 at different temperatures. The thicknesses of the oxide layers are 126.95 ± 2.00 µm, 67.50 ± 0.74 µm, 38.03 ± 0.82 µm and 9.99 ± 0.17 µm at 800 °C, 700 °C, 600 °C and 500 °C with 350 ppm hydrogen content, respectively. The future study for Zircaloy-4 will be focus on comparison of the oxidation law, the oxide-layer element compositions at different temperature and the hydrogen content consistent. 

## Figures and Tables

**Figure 1 materials-15-02997-f001:**
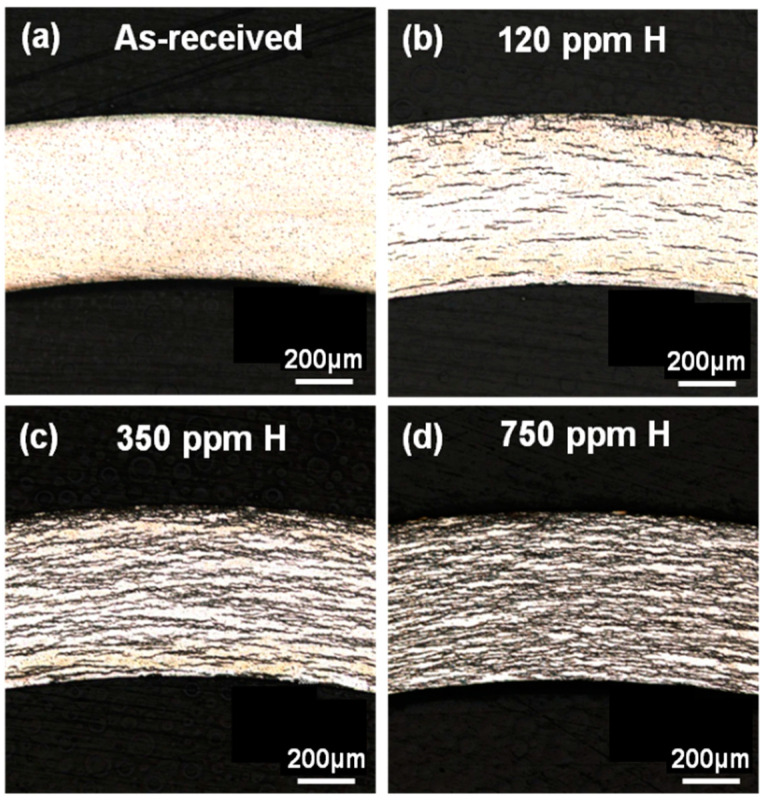
Optical micrographs of the (**a**) as-received Zircaloy-4 cladding, (**b**) Zircaloy-4 cladding with 120 ppm of H, (**c**) Zircaloy-4 cladding with 350 ppm of H, and (**d**) Zircaloy-4 cladding with 750 ppm of H.

**Figure 2 materials-15-02997-f002:**
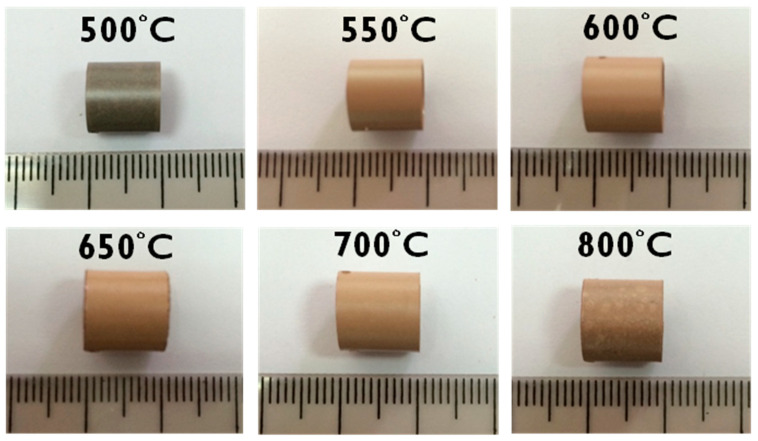
Surface features of the as-received Zircaloy-4 cladding specimens after the oxidation tests. (Time for testing: 500 °C for 150 h, 550 °C for 96 h, 600 °C for 50 h, 650 °C for 50 h, 700 °C for 10 h, and 800 °C for 3 h.).

**Figure 3 materials-15-02997-f003:**
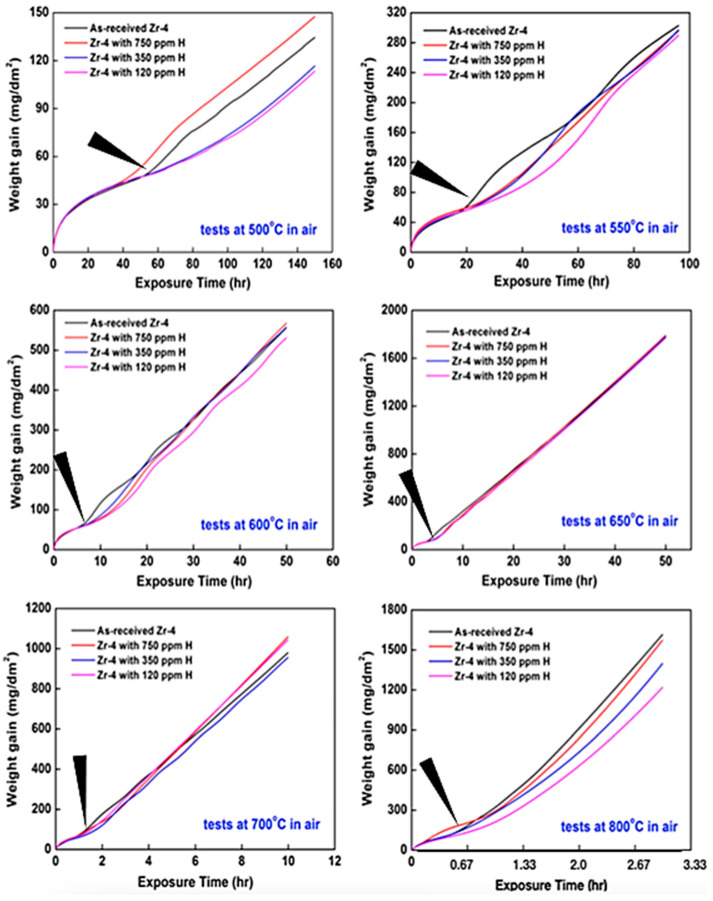
TGA results for the isothermal oxidation of the as-received and hydrided Zircaloy-4 cladding specimens in the temperature range of 500–800 °C (plot of weight gain per surface area vs. time).

**Figure 4 materials-15-02997-f004:**
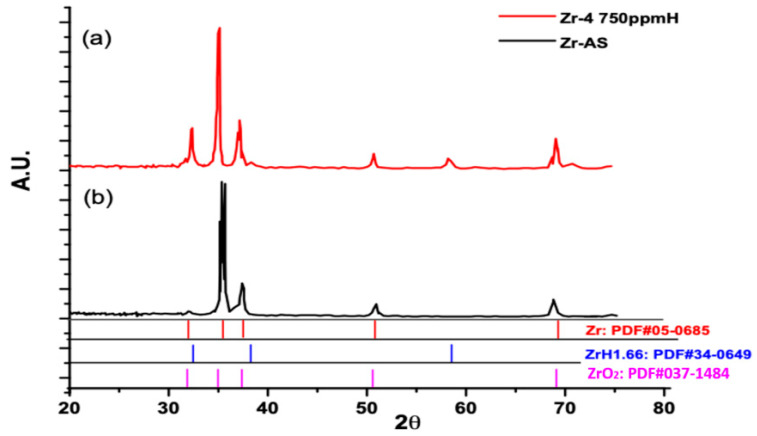
XRD characterization of (**a**) the unoxidized Zircaloy-4 specimen with 750 ppm of H and (**b**) the as-received Zircaloy-4 specimen.

**Figure 5 materials-15-02997-f005:**
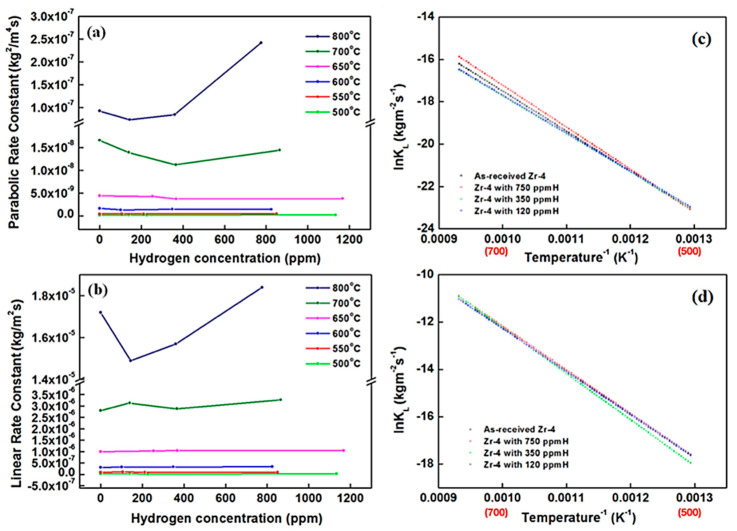
Variations in the (**a**) parabolic and (**b**) linear rate constants with the hydrogen concentration of the oxidized Zircaloy-4 specimens as well as the Arrhenius plots for the (**c**) pre-transition kinetics and (**d**) post-transition kinetics at 500–800 °C (in red) in air.

**Figure 6 materials-15-02997-f006:**
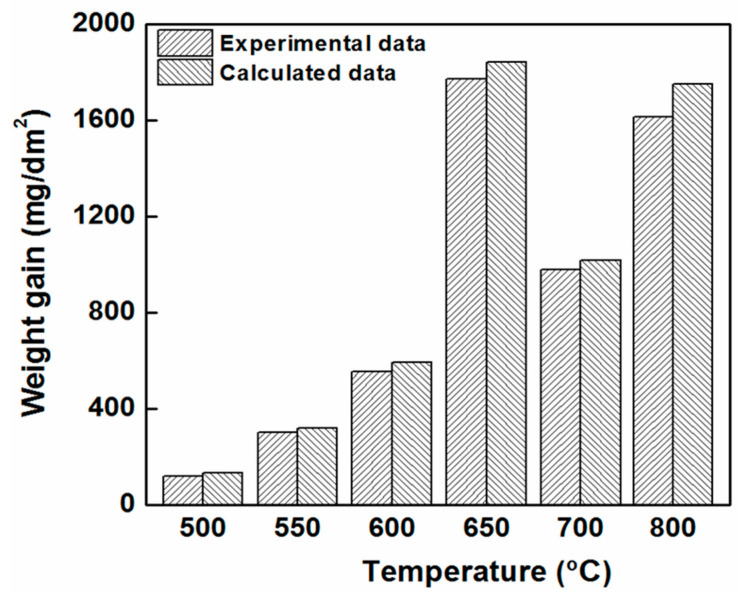
Experimental and calculated weight gain (∆*W*) data obtained from the oxidation rates determined for the as-received Zircaloy-4 cladding. The experimental and calculated results were determined at oxidation temperatures of 500, 550, 600, 650, 700 and 800 °C for exposure times of 150, 96, 50, 50, 10, and 3 h, respectively.

**Figure 7 materials-15-02997-f007:**
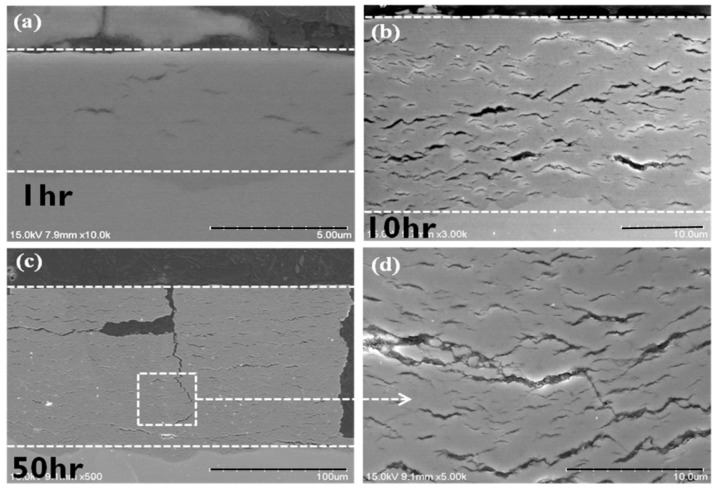
FE-SEM micrographs of the as-received Zircaloy-4 cladding specimens oxidized at 650 °C for (**a**) 1, (**b**) 10, and (**c**) 50 h. (**d**) Magnified image of the box in (**c**) for clearly illustrating the features of the oxide scale.

**Figure 8 materials-15-02997-f008:**
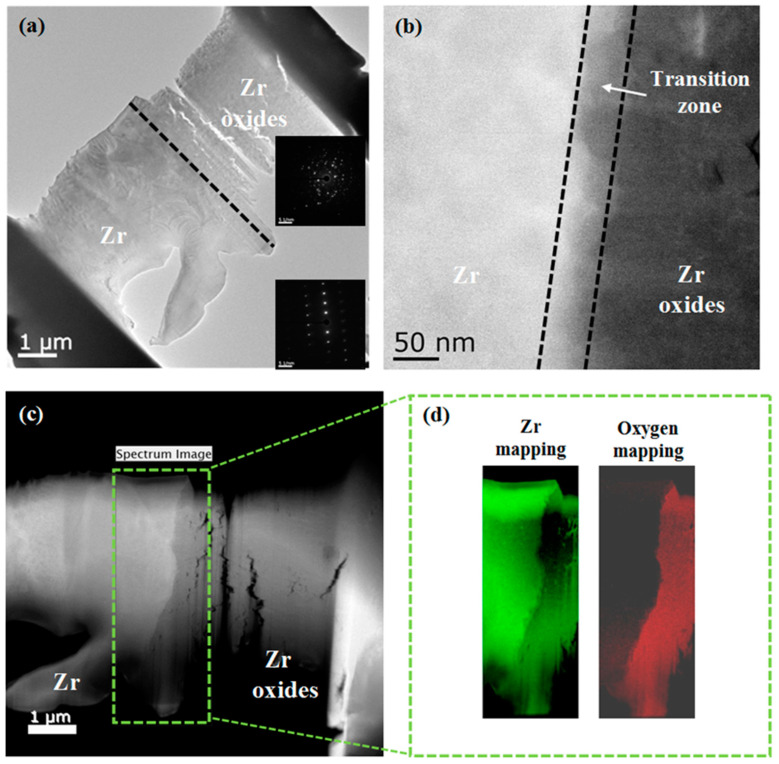
TEM micrographs of cross sections of the Zicroly-4 specimen with 350 ppm of hydrogen after oxidation at 650 °C for 25 h. (**a**) TEM image of the Zr substrate and oxides layers. The upper and lower inserts display the selected area electron diffraction (SEAD) patterns of the Zr oxides and substrate, respectively. (**b**) Bright-field image for examining the boundary between the Zr substrate and Zr oxides, (**c**) Scanning transmission electron microscopy (STEM) image used for EELS analysis, and (**d**) EELS mapping of Zr and O.

**Table 1 materials-15-02997-t001:** Composition and dimensions of the Zircaloy-4 cladding.

Element (wt%)	Sn	Fe	Cr	O	Zr	Dimensions
Zircaloy-4	1.2	0.21	0.11	0.12	balance	OD = 9.5 mm ID = 8.36 mm

**Table 2 materials-15-02997-t002:** Oxidation rate constants for the Zircaloy-4 cladding in the parabolic and linear rate regions.

Temperature (°C)	H Content		Rate Constant	
	Nominal (ppm)	Analyzed (ppm)	Parabolic Region (kg^2^/m^4^·s)	Linear Region (kg/m^2^·s)
500	unhydrided	unhydrided	1.02 × 10^−10^	2.33 × 10^−8^
	120	140	1.23 × 10^−10^	2.03 × 10^−8^
	350	227	1.25 × 10^−10^	1.21 × 10^−8^
	750	1133	1.69 × 10^−10^	2.41 × 10^−8^
550	unhydrided	unhydrided	4.20 × 10^−10^	9.48 × 10^−8^
	120	107	4.28 × 10^−10^	1.08 × 10^−7^
	350	212	4.16 × 10^−10^	9.16 × 10^−8^
	750	851	4.28 × 10^−10^	9.51 × 10^−8^
600	unhydrided	unhydrided	1.62 × 10^−9^	3.07 × 10^−7^
	120	101	1.30 × 10^−9^	3.19 × 10^−7^
	350	349	1.42 × 10^−9^	3.27 × 10^−7^
	750	825	1.40 × 10^−9^	3.38 × 10^−7^
650	unhydrided	unhydrided	4.38 × 10^−9^	1.00 × 10^−6^
	120	254	4.28 × 10^−9^	1.03 × 10^−6^
	350	368	3.72 × 10^−9^	1.04 × 10^−6^
	750	1167	3.77 × 10^−9^	1.04 × 10^−6^
700	unhydrided	unhydrided	1.66 × 10^−8^	2.80 × 10^−6^
	120	140	1.39 × 10^−8^	3.13 × 10^−6^
	350	366	1.12 × 10^−8^	2.88 × 10^−6^
	750	865	1.44 × 10^−8^	3.27 × 10^−6^
800	unhydrided	unhydrided	9.24 × 10^−8^	1.72 × 10^−5^
	120	144	7.29 × 10^−8^	1.49 × 10^−5^
	350	361	8.41 × 10^−8^	1.57 × 10^−5^
	750	776	2.42 × 10^−7^	1.84 × 10^−5^

**Table 3 materials-15-02997-t003:** Rate constant equations for the Zircaloy-4 cladding oxidized in air at temperatures between 500 and 800 °C.

Specimen	Rate Constant of Parabolic Region (kg^2^/m^4^·s)	Rate Constant of Linear Region (kg/m^2^·s)
As-received	*K_p_* = 4.06 × exp(−18,904/*T*)	*K_L_* = 391.51 × exp(−18,240/*T*)
750 ppmH	*K_p_* = 12.81 × exp(−19,778/*T*)	*K_L_* = 533.79 × exp(−18,456/*T*)
350 ppmH	*K_p_* = 1.16 × exp(−17,870/*T*)	*K_L_* = 1394.09 × exp(−19,473.27/*T*)
120 ppmH	*K_p_* = 1.21 × exp(−17,887/*T*)	*K_L_* = 368.71 × exp(−18,179.177/*T*)

Unit of *T*: Kelvin.

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
