# Peer review of "Oxidation Behavior Characterization of Zircaloy-4 Cladding with Different Hydrogen Concentrations at 500–800 °C in an Ambient Atmosphere"

_materials, 2022, doi:10.3390/ma15092997_

Round 1

Reviewer 1 Report

The authors investigated the oxidation behavior of pre-hydrided Zry-4 in air in the medium temperature range of 500-800°C. The are right that this temperature range is only rarely investigated; and also the effect of hydrogen (simulating high burnup) is of interest.

The manuscript has a number of significant drawbacks and should be strongly revised before being accepted for publication. In the following, only the major shortcomings are discussed, more comments and corrections can be found in the attached PDF file.

  1. The literature cited in the introduction and elsewhere looks like randomly collected and applied. E.g., for the description of air ingress scenarios, an overview paper on Arc welding of Zr alloys is cited, even a number of papers and reports on the topic itself are available.
  2. The TG results are presented systematically, but the post-test examinations not. Only one example of SEM examinations (as-received ZRY-4 at 650°C) and one example of TEM results (350 ppm H at 650°C) are given without any indication about the results for other samples.
  3. The TG results are simply described with words (larger/smaller as...), but no ideas are provided about possible mechanistic effects of H on the oxidation kinetics. E.g., local oxygen activities may be affected by H at the metal/oxide interface, which could affect oxidation and ZrN formation.
  4. Finally, it remains completely unclear, if there is an effect of H on the oxidation mechanism and kinetics. In section 3, you state "Thus , the oxidation behavior of Zircaloy 4 may be unaffected by hydrogen concentration at 500-800 °C"; and in the conclusion you write "The rate constant data indicated that the hydrogen concentration of the  Zircaloy-4 (in the range of 100–1000 ppm) significantly affected the pre-transition  oxidation in air at 500–800 °C"
  5. It is known from literature (see   https://doi.org/10.1002/maco.201307078) that post-transition kinetics are much more scattering than pre-transition kinetics. This must be discussed to clarify, if your data are relevant for finding an effect of H on the oxidation. Your TG curves look very similar to the ones published by Lasserre et al. with 10x repeating air oxidation tests under IDENTICAL conditions. Furthermore, e.g. Grosse et al. have shown, that experimental boundary conditions like flow rates and concentrations have strong effect on oxidation in nitrogen containing gases like air. (Grosse, M., Steinbrueck, M., Maeng, Y., Sung, J. Influence of the steam and oxygen flow rate on the reaction of zirconium in Steam/Nitrogen and Oxygen/Nitrogen atmospheres (2016) International Congress on Advances in Nuclear Power Plants, ICAPP 2016, 3, pp. 2103-2111). Your results have to be discussed in the light of these findings.
  6. Summarizing, this manuscript could be accepted for publication after serious revision taking into account the comments provided above and in the PDF file. 

Author Response

Dear Reviewer: 

On behaviors of all authors. We did answer point by point and revised our manuscript. 

Please find the attached. Thank you very much!

Best regards,

YUNG, Tung-Yuan

Reviewer 2 Report

This manuscript deals with the topic that covers oxidation of pre-hydrided zircaloy-4. Although there are several papers which also deal with this topic, this field is still interesting, and more studies need to be presented. It appears that the experiments described in this manuscript were well performed, however, more analysis and discussion need to be included in the manuscript. In addition, there are many errors (format, English, description), these errors must be fixed before the publication. The publication is not possible with current form, many revisions should be made. Detailed comments are shown below.

  1. At the introduction, following description should be fixed. “In nuclear power plants, the oxidation of fuel rods exposed to air can cause various accidents, such as the breach of a reactor pressure vessel or the loss of spent fuel pool cooling water[i].” It is wrong, excessive oxidation occurs as a result of an accident.

  1. At the introduction, more explanation (history, mechanism, related regulations) about hydrogen needs to be included in detail. Ex) hydrogen absorption process, effect of hydrogen on cladding integrity

  1. At the introduction, how you determine short or long term oxidation? Approximate scale or more explanation should be provided.

  1. More review of related previous studies should be presented at the introduction. It is important to emphasize what was already discovered and what was not found by previous studies.

  1. At the introduction, there is the following description. “Few studies have investigated the long-term oxidation behaviors of Zircaloy cladding in air at temperatures below 700 °C”. Does 700 °C have any special meaning? Why is it important to investigate oxidation at this temperature range?

  1. Formats must be fixed.
    Ex) figure captions (fig. and figure), references ([5]), and so on.

  1. Time scales should be presented at the caption of figure 2.

  1. There is the following description in the manuscript. “Moreover, in terms of their shade of brown, variations were observed the surfaces of the cladding specimens oxidized between 500 and 800°C”. It is hard to understand what variation is. More explanation is necessary.

  1. Why the results of Baek et al. were mentioned? Detailed comparison is necessary.

  1. It is better to include general scale in Fig 5 (c) and (d).

  1. It is well known that hydrogen absorption to zirconium alloy cladding is not homogeneous. Distribution of hydrogen within zirconium alloys should be discussed. It is better to include some data showing the distribution of hydrogen.

  1. In manuscript, there is the following description. “(as indicated by the arrows in Fig. 4)”. There is no arrow in Fig. 4.

  1. The discussion about oxygen stabilized alpha zirconium needs to be revised. Oxygen stabilized alpha zirconium is important at temperatures where zirconium substrate is turned to beta phase. However, at temperatures where authors experimented the fraction of beta phase zirconium may be small (there may be some beta phase zirconium at 800 °C.
    In addition, it is hard to understand why cracks are formed as oxygen stabilized alpha zirconium is oxidized, because there are many contradicting experimental results.

  1. In the manuscript, there is a following description about zirconium nitride. “zirconium nitride because the thicknesses of the transition layer and transition zone were approximately 25–30 and 50 nm, respectively” It is hard to understand. Why zirconium nitride was not detected during TEM and XRD analysis?

  1. Conclusions should be revised. Conclusion should put more emphasis on the new findings presented in the manuscript.

Author Response

(The authors gave the same response as above.)

Reviewer 3 Report

The article is devoted to the study of the degradation processes of Zircaloy-4 as a result of hydrogen saturation. In general, this direction is of scientific interest and practical importance for the direction of materials science. According to the reviewer, this work can be accepted for publication after the authors answer a number of questions that arose when reading the article.

1 Authors should clarify the relevance and purpose of the study.
2 The authors should explain in the experimental part the choice of temperatures for conducting experiments.
3 The authors should give explanations regarding the data shown in Figure 6, which is associated with such a decrease in weight gain at 700 ° C.
4 It can be seen from the X-ray data that the hydrogenation processes lead to a change in the phase composition, but the authors should give the numerical value of the phase content.
5 The authors are invited to amend Figure 3 to make it comparative, it is necessary to bring all scales to the same scale for ease of comparison.
6 The conclusion requires revision, with the addition of the main results, as well as the definition of plans for future research.

Author Response

(The authors gave the same response as above.)

Round 2

Reviewer 1 Report

Unfortunately, the revised version of the manuscript does not really provide major improvements. The main issues are:

1) The English, especially of the new parts, is not acceptable. Many sentences are hard to understand or mix cause and effect.

- e.g., your first sentence "... oxidation cause accidents, ..." is wrong. It is the other way round: severe oxidation of Zry cladding is caused by various accident scenarios.

2) The many comments and corrections in the PDF file provided with the first review have not taken into account at all.

3) This includes e.g. suggested references. The new reference Grosse et al. does not appear in the list of references. Important literature on air oxidation of Zr alloys by Duriez, Steinbrueck, and Lasserre is missing. It should be read, understood and cited. 

4) Finally, it is still not clear if the observed effects are due to hydrogen or due to the natural scattering of oxidation kinetics especially after transition.

As already stated in the first review, the results obtained have the potential for a good paper. But the manuscript should be revised in terms of content and language by a senior scientist. The obtained results must be clearly stated and  compared to the available international literature. 

Author Response

1) The English, especially of the new parts, is not acceptable. Many sentences are hard to understand or mix cause and effect.

- e.g., your first sentence "... oxidation cause accidents, ..." is wrong. It is the other way round: severe oxidation of Zry cladding is caused by various accident scenarios.

Response to reviewer: We modified the above sentence as following : ” In nuclear power plants, the excessive oxidation of fuel rods exposed to air and steam atmospheres may cause serious accidents, such as the breach of a reactor pressure vessel or the loss of spent fuel pool cooling water.”

2) The many comments and corrections in the PDF file provided with the first review have not taken into account at all.

Response to reviewer: We lost to check your PDF file first revision. We answered as following:

  1. This is not a good reference at this place, it is just secondary literature on another topic.

Use e.g. IAEA-TECDOC-1949 - Phenomenology, Simulation and Modelling of Accidents in Spent Fuel Pools (https://www.researchgate.net/publication/353716318_IAEA-TECDOC-1949_-_Phenomenology_Simulation_and_Modelling_of_Accidents_in_Spent_Fuel_Pools#fullTextFileContent)

Response to reviewer: We added this reference in reference [1]. Thank you for your suggestion.

  1. Maximum temperatures in dry storage canisters are <400°C!

Response to reviewer: Thank you for your reminding. We wrote this sentence to enhance the cladding integrity is important.

  1. Depending on concentration and temperature, hydrogen is either dissolved or precipitated as hydride. This has to be discussed for the conditions investigated in this study.

Response to reviewer: In this study, we simulated the after-burned fuel rod cladding with hydrogen charging. In the experimental section, we dressed the hydrogen charging as “To simulate the properties of spent fuel cladding, the Zircaloy-4 cladding tubes were first hydrogen-charged to the target hydrogen concentrations of 120, 350, and 750 ppm”. We added the following sentence in the Introduction section as: “As the spent fuel zirconium claddings after burn-up, the 100–1000 ppm hydrogen contents are observed.”

  1. TGA is which type? I

Response to reviewer: We added the type in experimental section. Setsys EVO, Setaram Inc.

  1. This is the only reaction for Zr oxide formation. Zr dissolves a certain amount of O (according to the Zr-O phase diagram).

Response to reviewer: Thank you for your comments. We added one more reaction formula to form ZrO2 in the revision manuscript. “Zr + 2H2O → ZrO2 + H2,” The experimental environment is in ambient condition which could contain moisture.

  1. There could be NO ZrN in Zr-H before oxidation.

Response to reviewer: Thank you for your comment. In figure 4, there is no significant Zr-N peaks as-received Zr-4. It is ambiguously identifying with Zr-H and Zr-N.

  1. A very good overview paper on Zry oxidation was published by Schanz et al. (doi:10.1016/j.nucengdes.2004.02.013).

Response to reviewer: We added this reference in reference [28].

A recent overview paper was published by Steinbrueck (https://doi.org/10.1016/B978-0-12-819726-4.00006-5).

Response to reviewer: Thank you for your kindly comment. We added this in reference [5].

  1. These statements make no sense and are not understandable. Please modify and explain what you are really want to say.
  2. No reference [5] available in the list.

Response to reviewer: We added two papers in reference [5]. Thank you for your help.

  1. This is a commonplace and true for all diffusive driven processes in nature.

Response to reviewer: Thank you for your comment. We deleted this sentence in latest manuscript.

  1. J should be used as unit here and throughout the manuscript.

Response to reviewer: Thank you for your comment. We change the constant as 8.314J/mol.

  1. one decimal place should enough for these and the following data.

Response to reviewer: Thank you for your comment. We change these data with one decimal.

  1. calculated based on the Arrhenius equations determined in this study? Then, is is normal that you get a good correlation.

Response to reviewer: Thank you for your comment. It is true as the Arrhenius equation is quit good fitting results in this study.  

  1. In recent studies, it was shown that nitrogen just acts as a kind of "catalyst" due to TEMPORARY formation and re-oxidation of ZrN. That means, that N not necessarily must be found even is may have affected the oxidation mechanisms.

Response to reviewer: Thank you for your help. We modified the sentence in the latest manuscript as following: “There is a report for nitride acts as the “catalyst” in zirconium oxidation [5].”

  1. See comment and suggestion above. Here you mix statements on the tests conditions and results.

Response to reviewer: Thank you for your help. We are happy with your comments to make our manuscript better and better.

3) This includes e.g. suggested references. The new reference Grosse et al. does not appear in the list of references. Important literature on air oxidation of Zr alloys by Duriez, Steinbrueck, and Lasserre is missing. It should be read, understood and cited. 

Response to reviewer: Thank you for your kindly comments.

4) Finally, it is still not clear if the observed effects are due to hydrogen or due to the natural scattering of oxidation kinetics especially after transition.

Response to reviewer: Thank you for your comments. We plan to do more detail study in the future to reveal the hydrogen content effect the oxidation kinetics or not.

As already stated in the first review, the results obtained have the potential for a good paper. But the manuscript should be revised in terms of content and language by a senior scientist. The obtained results must be clearly stated and compared to the available international literature. 

Response to reviewer: We thank you for your kindly comments with our manuscript. Hopefully, the latest manuscript could be fit as your high standard.

Reviewer 2 Report

The authors adequately modified the manuscript following the revisions.

Author Response

Response to Reviewer: Thank you for your comments. 

Reviewer 3 Report

The authors answered all the questions posed. The article may be accepted for publication.

Author Response

Response to reviewer: Thank you for your comments.

Round 3

Reviewer 1 Report

Dear authors, the manuscript has been improved to some extent, but still contains errors in content and language. Please see the attached PDF file with corrections and comments. 

Author Response

Response to reviewer: 

On behavior of authors, we are sincerely thank to the reviewer's efforts to make our manuscript better and better.   

  1. Rather: In nuclear power plant, severe accidents, such as..., result in exposure of fuel rods to high-temperature steam and air with excessive oxidation of the Zr alloy fuel cladding.  //Response:: Thank you for your modification. Please find the revision manuscript. 
  2. Oxidation in air van only result in Equ.1 and 
    Zr + 0.5 N2  ZrN
    Equ.2 only if air is humid, this should be mentioned. 
    There are not "many possible other reactions" //Response: Thank you for comments. We added one sentence for Equ. 2 in the moisture air. Please find the latest revision.
  3. Reference is missing. //Response: Thank you for carefully check our manuscript. We added in reference [5].
  4. How do you explain the ZrN peaks before the oxidation experiments in air?
    Just seen in Fig.4: PDF40-1275 is not Zr nitride, but N-stabilized alpha-Zr(N). This may be very similar to alpha-Zr(O), which would be much more reasonable. PLease check. //Response: Thank you for your help. Yes, your comment is correct with comparison of TEM results. Since the Zr(O) and ZrN are similar peaks in PDF database. We modification the Fig. 4 and the description in the latest revision. 
  5. Please see Lasserre's Paper, which shows that such behavior is also observed at Zry without H pre-loading. Fig.3 in DOI: 10.1002/maco.201307078 // Response: We added this in reference [23]. Thank you for help.
  6. This is wrong.
    During the pre-transition period, a protective ZrO2 scale is formed (and oxidation determined by diffusion of O through this growing scale). Various mechanics may result in the formation of cracks (breakaway). After transition, the oxidation kinetics is determined by the diffusion of the oxidizing gas through these cracks to the M/O interface. // Response: We thank you for your correction. We modified in our manuscript. 
  7. Refernce [xx] is missing. // We added in reference [29]. Thank you for your help. 
  8.   ??// "be effected". Thank you for comment. 
  9.  Looks like two references// Response: Thank you for your comment. We added the similar two references in reference [5], [13] and [28].
  10. Thank you for your help to correct our manuscript in English. 

     Best regards, 

Tung-Yuan Yung